# Comparing Survival after Resection of Pancreatic Cancer with and without Pancreatic Cysts: Nationwide Registry-Based Study

**DOI:** 10.3390/cancers14174228

**Published:** 2022-08-30

**Authors:** Myrte Gorris, Nadine C. M. van Huijgevoort, Arantza Farina, Lodewijk A. A. Brosens, Hjalmar C. van Santvoort, Bas Groot Koerkamp, Marco J. Bruno, Marc G. Besselink, Jeanin E. van Hooft

**Affiliations:** 1Amsterdam UMC, Location University of Amsterdam, Department of Gastroenterology and Hepatology, 1081 HV Amsterdam, The Netherlands; 2Amsterdam Gastroenterology Endocrinology Metabolism, 1105 BK Amsterdam, The Netherlands; 3Amsterdam UMC, Location University of Amsterdam, Department of Surgery, 1105 AZ Amsterdam, The Netherlands; 4Cancer Center Amsterdam, 1081 HV Amsterdam, The Netherlands; 5Amsterdam UMC, Location University of Amsterdam, Department of Pathology, 1081 HV Amsterdam, The Netherlands; 6Department of Pathology, University Medical Centre Utrecht, Utrecht University, 3508 GA Utrecht, The Netherlands; 7Department of Surgery, St. Antonius Hospital, 3435 CM Nieuwegein, The Netherlands; 8Department of Surgery, University Medical Centre Utrecht, Heidelberglaan 100, 3584 CX Utrecht, The Netherlands; 9Department of Surgery, Erasmus MC Cancer Institute, University Medical Centre Rotterdam, Doctor Molewaterplein 40, 3015 GD Rotterdam, The Netherlands; 10Department of Gastroenterology and Hepatology, Erasmus MC Cancer Institute, University Medical Centre Rotterdam, Doctor Molewaterplein 40, 3015 GD Rotterdam, The Netherlands; 11Department of Gastroenterology and Hepatology, Leiden University Medical Center, Albinusdreef 2, 2333 ZA Leiden, The Netherlands

**Keywords:** pancreatic neoplasms, pancreatic cyst, surgical oncology, survival analyses, Kaplan–Meier estimates

## Abstract

**Simple Summary:**

Pancreatic cancer has a poor prognosis, even in patients that can be surgically treated with curative intent. An interesting subgroup of resected pancreatic cancers are those associated with pancreatic cystic neoplasms (PCN), since overall survival might differ from pancreatic cancer not associated with PCN. Although several single-center studies published conflicting data on this topic, nationwide studies are lacking. In this nationwide, registry-based study, we aimed to compare the overall survival between patients with PCN-associated pancreatic cancers to those with pancreatic cancer not associated with PCN. We found that 12% of resected pancreatic cancers patients were PCN-associated. Overall survival was better in patients with PCN-associated pancreatic cancer as compared to those not associated with PCN. Future prospective studies should focus on the impact of these findings, such as the impact of (neo)adjuvant treatment regimens in this specific patient group.

**Abstract:**

Background: Outcome after resection of pancreatic ductal adenocarcinoma associated with pancreatic cystic neoplasms (PCN-PDAC) might differ from PDAC not associated with PCN. This nationwide, registry-based study aimed to compare the overall survival (OS) in these patients. Methods: Data from consecutive patients after pancreatic resection for PDAC between 2013 and 2018 were matched with the corresponding pathology reports. Primary outcome was OS for PCN-PDAC and PDAC including 1-year and 5-year OS. Cox regression analysis was used to correct for prognostic factors (e.g., pT-stage, pN-stage, and vascular invasion). Results: In total, 1994 patients underwent resection for PDAC including 233 (12%) with PCN-PDAC. Median estimated OS was better in patients with PCN-PDAC (34.5 months [95%CI 25.6 to 43.5]) as compared to PDAC not associated with PCN (18.2 months [95%CI 17.3 to 19.2]; hazard ratio 0.53 [95%CI 0.44–0.63]; *p* < 0.001). The difference in OS remained after correction for prognostic factors (adjusted hazard ratio 1.58 [95%CI 1.32−1.90]; *p* < 0.001). Conclusions: This nationwide registry-based study showed that 12% of resected PDAC were PCN-associated. Patients with PCN-PDAC had better OS as compared to PDAC not associated with PCN.

## 1. Introduction

Pancreatic ductal adenocarcinoma (PDAC) is projected to become the second most common cause of cancer-related death in 2030 [1]. Currently, surgical resection in combination with systemic therapy is the only treatment option for long-term survival, with a 5-year survival rate of around 20% [2,3].

An interesting subgroup of resected PDAC is those associated with pancreatic cystic neoplasms (PCN), such as intraductal papillary mucinous neoplasms (IPMN) and mucinous cystic neoplasms (MCN). PCN may be related to PDAC in multiple ways. First, PCN are premalignant and thus may be a direct precursor lesion for PDAC. The annual risk of progression to malignancy depends on the type of PCN and increases when worrisome features and/or high-risk stigmata are present [4]. Second, concomitant PDAC is more often detected in patients with PCN. These two can often not be distinguished from one another in the surgical specimen without additional molecular analysis of clonality [5]. Therefore, we will define both as ‘PCN-associated PDAC’ (PCN-PDAC).

It is currently unclear whether the prognosis of patients with PCN-PDAC after surgical resection differs from other patients with PDAC. A single-center study from Sweden including 122 patients with IPMN-PDAC and 411 other patients with PDAC found a more favorable 2-year median overall survival (OS) for patients with IPMN-PDAC after surgical resection (33.6 months vs. 19.3 months, *p* = 0.001) [6]. In multivariable Cox regression analysis, however, PCN-PDAC was not a predictor for longer survival. In addition, a multicenter Japanese study showed a longer median OS in patients with IPMN-PDAC after surgical resection compared to patients with other PDAC (46 months vs. 12 months, *p* < 0.001) [7]. Nationwide studies on this topic are, however, lacking. The aim of this nationwide, registry-based study was therefore to investigate OS after resection of PCN-PDAC versus PDAC not associated with PCN.

## 2. Materials and Methods

### 2.1. Study Design

All patients who underwent pancreatic resection between 2013 and 2018 with pancreatic ductal adenocarcinoma in the resection specimen were included from the Netherlands Cancer Registry (NCR). The nationwide NCR records data on all patients with newly diagnosed cancer in the Netherlands (a country with nearly 17.3 million inhabitants). The NCR is based on notification of all newly diagnosed malignancies by the nationwide network and registry of histo- and cytopathology in the Netherlands (PALGA) database, which covers all pathology reports [8]. The NCR is supplemented with data from the National Registry of Hospital Discharge Diagnoses. Completeness is estimated to be at least 95%. Trained registrars routinely collect data on patient characteristics, tumor type, and primary cancer treatment (tumor resection, radiotherapy, chemotherapy) extracted from medical records in all Dutch hospitals. Actual vital status (dead or alive) was routinely obtained until February 2021 from the Municipal Personal Records Database, which contains information on the vital status of all Dutch inhabitants. This study was conducted in accordance with the STROBE guidelines for reporting observational studies and the study proposal was approved by the review board of the Netherlands Comprehensive Cancer Organization (IKNL), the scientific council and privacy committee of PALGA, and the scientific committee of the Dutch Pancreatic Cancer Group (DPCG) [9]. The study was conducted according to the guidelines of the Declaration of Helsinki and approved by the institutional review board of the Amsterdam UMC on 13 December 2019 (study number W19_465). Patient consent was waived due to the use of an anonymized study database.

### 2.2. Study Population

All patients with International Classification of Diseases for Oncology (ICD-O) morphology code ‘Pancreas’ (C25) and tumor classification ‘ductal adenocarcinoma’ according to the World Health Organization (WHO) classification were selected from the NCR [10,11]. Patients diagnosed with PDAC during autopsy and patients aged younger than 18 years at diagnosis were excluded. Subsequently, patients from the NCR database were matched with the corresponding pathology reports from the PALGA database. After matching, exclusion criteria were: resection specimen report not available, non-reliable database matching between NCR and PALGA databases, patient missing in PALGA database, other malignancies in the resection specimen (e.g., primary cholangiocarcinoma, pancreatic neuroendocrine tumor), benign disease in the resection specimen, insufficient data in the pathology report, and re-resections.

### 2.3. Data Collection

Pathology reports, including radicality margins, were evaluated by the coordinating investigators (NvH and MG). In case of inconclusive reports, a dedicated pancreatobiliary pathologist (AF) and/or pancreatobiliary surgeon (MGB) was consulted. Pathology findings were reported according to a national standardized protocol from 2016 onwards. Preoperative imaging characteristics were not available in the nationwide registry. Thus, all patients included in this study were diagnosed with PCN-PDAC based on the findings in the resection specimen. We defined PCN-PDAC based on the microscopic and conclusion section of the pathology report as: (a) PDAC directly originating from PCN based on the relationship of the two conditions, (b) PDAC concomitant with PCN elsewhere in the pancreas, or (c) PDAC and PCN with an indeterminate relationship. PCN compromised both IPMN and MCN, which were defined according to the fourth edition of the World Health Organization (WHO) Classification of Tumours of the Digestive System [11,12]. Pathology reports describing PDAC lesions without the description of any PCN component were classified as PDAC not associated with PCN. pTNM-stage, pT-stage, and pN-stage were based on the pathological tumor-node-metastasis (TNM) classification at the time of registration (seventh edition of IUCC TNM staging during 2010–2017, eighth edition from 2018 onwards) [13,14]. Tumor size (pT-stage) was based on the size of the invasive component. Microscopic radicality (R0/1) was defined according to the International Collaboration on Cancer Reporting [15].

### 2.4. Outcomes

The primary outcome was OS in PCN-PDAC and PDAC not associated with PCN (including 1-year and 5-year OS) and was defined as the time between date of diagnosis and date of death or censored at last follow-up. Follow-up information was obtained by linkage of the NCR with the Municipal Personal Records Database (updated in February 2021). This unique nationwide coverage enables an accurate estimation of the OS in PCN-PDAC and PDAC in the Netherlands.

### 2.5. Statistical Analysis

Missing data were handled using multiple imputation with predictive mean matching, in which 10 imputation dummy sets were created. Baseline characteristics were characterized by descriptive statistics, presented as a mean with standard deviation (SD) for normally distributed continuous data and as a percentage for categorical data. Comparisons were made using Chi-square statistics (or Fisher’s exact test where appropriate) for numerical data, and the t-test was used for continuous variables. OS was calculated using the Kaplan–Meier method and compared using the log-rank test. To demonstrate whether PCN-PDAC was associated with overall survival, a multivariable Cox regression model was computed to correct for prognostic factors. The proportional hazard assumption was verified by establishing log-minus-log survival plots for every covariate in the Cox regression analysis. Potential confounders were sex, age, WHO score, American Society of Anesthesiologists (ASA) score, and tumor characteristics (i.e., pT-stage, pN-stage, perineural invasion, vascular invasion, R0/1 resection, and differentiation grade). All variables showing a statistically significant association with OS in univariate Cox regression analysis were included in the multivariable model. Results were presented as hazard ratios (HR) with 95% confidence intervals (CI). All *p*-values were based on a 2-sided test, and *p*-values < 0.05 were considered statistically significant. Data were analyzed using IBM SPSS Statistics for Windows, version 28 (IBM Corp., Armong, NY, USA).

## 3. Results

### 3.1. Study Cohort

Overall, we included 1994 patients after resection of PDAC (Figure 1). This included 233 patients (12%) with PCN-PDAC and 1761 patients (88%) with PDAC not associated with PCN. IPMN-associated PDAC was reported in 222/233 patients (95%) and MCN-associated PDAC in 11/233 patients (5%). In the pathology reports, 172 (74%) of PCN-PDAC were reported as PDAC directly originating from PCN, 49 (21%) as concomitant with PCN elsewhere in the pancreas, and 12 (5%) as having an indeterminate relationship. The study group had a mean age of 67 years (SD ± 9.4), and 52% of patients were male. Fifty-five percent (*n* = 123) of the patients with PCN-PDAC received (neo)adjuvant therapy compared to 62% (*n* = 1084) of the patients with PDAC not associated with PCN. Both pT-stage and pN-stage were lower in PCN-PDAC patients (*p* < 0.001 for both, Table 1). Vascular and perineural invasions were detected less often in patients with PCN-PDAC (*n* = 1084 [62%] vs. *n* = 97 [43%], *p* < 0.001 and *n* = 1524 [86%] vs. *n* = 157 [70%], *p* < 0.001, respectively), whereas more patients with PCN-PDAC underwent R0 resection (*n* = 118 [52%] vs. *n* = 748 [42%], *p* = 0.007). Detailed tumor characteristics are depicted in Appendix A.

### 3.2. Overall Survival

After a median follow-up of 19.3 months, a total of 1614 patients had died. Of these, 141 (61%) patients with PCN-PDAC were deceased as compared to 1473 (84%) patients with PDAC not associated with PCN. The median estimated OS was better in patients with PCN-PDAC (34.5 months, 95%CI 25.6 to 43.5) as compared to PDAC not associated with PCN (18.2 months, 95%CI 17.3 to 19.2, HR 0.53; 95%CI 0.44–0.63, *p* < 0.001, Figure 2). The 1-year and 5-year estimated survival rates in PCN-PDAC patients were 80.7% (95%CI 75.0 to 85.2) and 34.0% (95%CI 26.7 to 41.4), respectively, as compared to 68.3% (95%CI 66.1 to 70.4) and 14.6% (95%CI 12.8 to 16.5), respectively, in patients with PDAC not associated with PCN. When corrected for pTNM-stage, the association between PCN-PDAC and OS remained (adjusted HR 0.60 [95%CI 0.50–0.71], *p* < 0.001). In addition, the association between PCN-PDAC and OS remained after correction for prognostic factors (adjusted HR 0.65 [95%CI 0.55–0.78]; *p* < 0.001, Table 2). Other prognostic factors that were associated with OS were pT-stage (HR 1.71 [95%CI 1.14–2.56] for T4 staging), pN-stage (HR 1.65 [95%CI 1.44–1.88]), perineural invasion (HR 1.42 [95%CI 1.19–1.70]), R0/1 resection (HR 1.25 [95%CI 1.13–1.39]), and differentiation grade (HR 1.28 [95%CI 1.07–1.52] for moderate differentiation and HR 1.83 [95%CI 1.51–2.21] for poor differentiation).

## 4. Discussion

This nationwide, registry-based study found better OS after resection for PCN-PDAC as compared to resection of PDAC not associated with PCN. This difference remained after correction for prognostic factors.

This is the first nationwide, registry-based study to compare oncological outcomes of resected PCN-PDAC to resected PDAC. Previous studies that compared OS between PCN-associated PDAC and PDAC not associated with PCN are mostly single-center, retrospective cohort studies. In line with our findings, a Japanese study from 2011 reported better median OS for IPMN-PDAC (46 months for PDAC derived from IPMN and 57 months for PDAC concomitant with IPMN, respectively) as compared to other PDAC patients (12 months), although regression analysis to correct for other prognostic factors was not performed [7]. In addition, Gavazzi et al. reported better OS in PCN-PDAC (*n* = 43, 65.4%) as compared to other PDAC (*n* = 289, 14.2%) [16]. A recent Korean paper also observed better 5-year OS in 67 IPMN-PDAC patients (66%) as compared to 551 other PDAC patients (14%, *p* < 0.001) [17]. Another recent study underlined these findings and found lower recurrence (*p* = 0.006) and death rates (*p* = 0.007) for 92 invasive IPMN patients as compared to 304 PDAC patients [18]. In a total of 330 patients, Capretti et al. reported higher median disease-free survival (60.3 months) based on 43 invasive-IPMN patients as compared to 287 PDAC patients (median 23.8 months, *p* < 0.001) [19]. These differences in OS might suggest that PCN-PDAC shows different tumor biology as compared to PDAC not associated with PCN. However, a recent study did not find differences in the alteration frequency of the major driving genes between PDAC and IPMN-associated PDAC [20]. This hypothesis needs to be verified in larger prospective cohorts, in which more extensive molecular characterization such as whole genome sequencing, mRNA sequencing, and even methylation analysis is performed on both PCN-PDAC and PDAC not associated with PCN tumors, to create insight in possible differences in the tumor genome.

In contrast, a recently published retrospective single-center study by Holmberg et al. reported that IPMN-PDAC was not associated with death in 513 patients (122 with IPMN-PDAC and 391 with other PDAC) [6]. Another single-center study also found no differences in overall and disease-free survival between PDAC concomitant with IPMN and other PDAC in 158 propensity score-matched subjects [20].

The reported proportion of PCN-PDAC among resected PDAC lesions varies widely. The aforementioned study by Holmberg et al. reported a proportion of 122 IPMN-PDAC among 513 PDAC patients (23%), twice higher than the 12% in our cohort [6]. Another study found a relatively low proportion of 2%, although the authors compared 7605 resected PDAC patients from a nationwide registry with 122 resected IPMN-PDAC patients from their own institution [7]. Gavazzi et al. diagnosed 43 patients with IPMN-PDAC among a cohort of 332 resected PDAC patients (13%) [16], whereas Marsoner et al. reported 30 (12%) resected IPMN-PDAC patients against 221 other PDAC resections [21]. However, all these results were obtained from single-center, retrospective cohort studies and might thus be influenced by local referral patterns.

The results of this study should be interpreted in light of some limitations. First, due to its retrospective nature, conclusions on causality cannot be drawn. The difference in OS might be biased by other factors (e.g., lead time bias) and might thus not entirely be attributed to a difference in tumor biology. However, we did correct for prognostic tumor characteristics (e.g., pT-stage and pN-stage) in our Cox regression analysis. Nevertheless, not all factors that influence OS (e.g., chemotherapy treatment) were included since the observed treatment effect might not be solely attributed to the given treatment (e.g., immortal time bias). In addition, data on other factors that might have influenced OS (e.g., presenting symptoms such as jaundice or cachexia, postoperative complications) were not available in our dataset. However, the results of this study were probably not significantly influenced by differences in postoperative complications, since in-hospital mortality in the Netherlands is 1.3% after distal pancreatectomy and 2.4% after Whipple resection [22]. Furthermore, disease-free intervals and recurrence patterns were not registered in the nationwide registry and could therefore not be analyzed. Second, the proportion of PCN-PDAC may be underreported in our study, since pathologists particularly report PDAC lesions which clearly derived from PCN and thus might not report smaller PCN or lesions without a clear association. Furthermore, data on the presence of concomitant PCN in the non-resected pancreas were not available. Third, the diagnosis of PCN-PDAC was based on the conclusion and microscopy section of the pathology reports. Data regarding radiological imaging, sampling of the resection specimen, the macroscopic description of the lesion, and corresponding pictures of the resection specimen were not available. In addition, a nationwide standardized reporting protocol for pathologists was only implemented from 2016 onwards, and the pathology reports often lacked a detailed description of the PCN component. As a consequence, we were unable to analyze detailed PCN characteristics (e.g., PCN size) and decided to analyze all patients in which a PCN lesion was mentioned as ‘PCN-associated PDAC’. Thus, possible differences between PCN concomitant with PDAC and PCN directly related to PDAC should be further investigated in future, prospective cohorts with standardized pathological slicing techniques and availability of preoperative radiological imaging. In addition, selection bias might have been introduced since we excluded unclear pathology reports. Lastly, OS might have been underestimated since time to surgical resection was not accounted for, although the OS in patients with PDAC not associated with PCN was comparable to the long-term follow-up data of the Dutch prospective PREOPANC trial [3].

Despite its limitations, this nationwide study consists of a large sample of patients with resected PDAC, thereby providing real-world data on the proportion and OS of PCN-PDAC. By matching the national PALGA database and national NCR database, all relevant clinical characteristics were available, and more importantly, the NCR provided unique, long-term follow-up data. Furthermore, we modelled missing data by using multiple imputation, thereby making our analyses less susceptible to reporting bias. Lastly, we corrected for prognostic factors that might have influenced the observed difference in OS by computing a multivariable Cox regression model.

## 5. Conclusions

In conclusion, this study found a 12% rate of PCN-PDAC among resected PDAC. Patients after resection of PCN-PDAC had longer OS as compared to PDAC not associated with PCN. Future prospective studies should determine the impact of these findings such as the impact on (neo)adjuvant treatment regimens in this specific patient group.

## Figures and Tables

**Figure 1 cancers-14-04228-f001:**
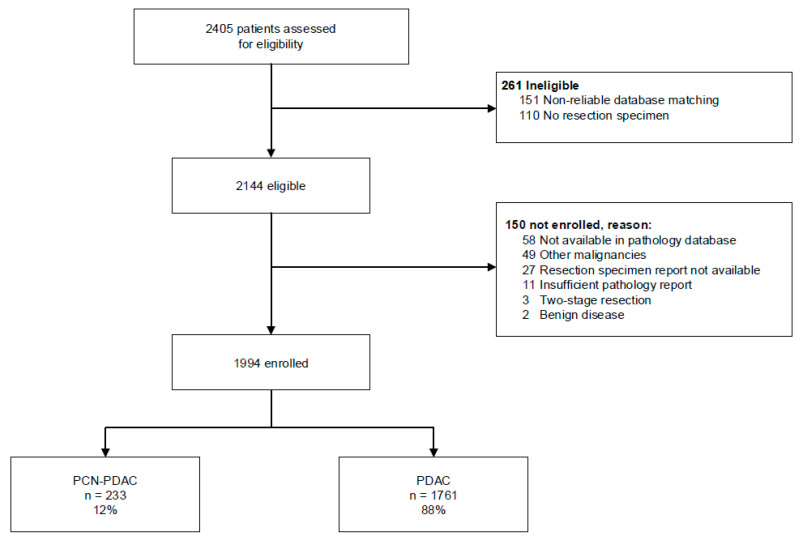
Study design. Abbreviations: PCN = pancreatic cystic neoplasms. PDAC = pancreatic ductal adenocarcinoma.

**Figure 2 cancers-14-04228-f002:**
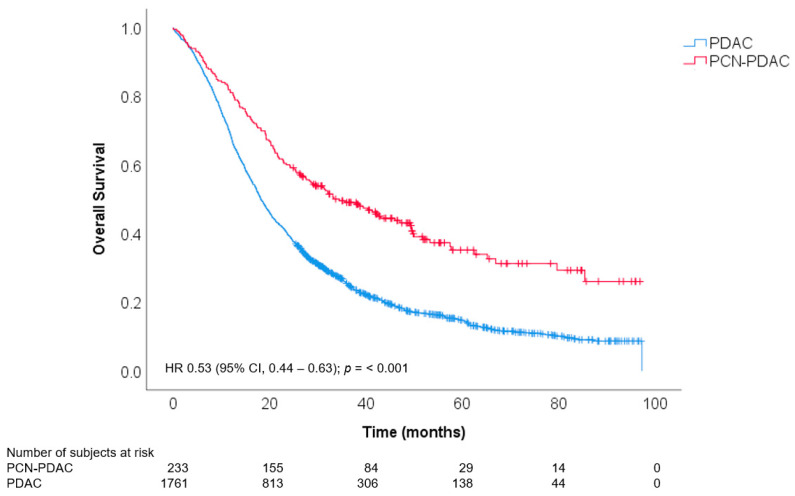
Kaplan–Meier estimates of overall survival after resection of PCN-PDAC versus PDAC not associated with PCN. Abbreviations: CI = confidence interval. HR = hazard ratio. PDAC = pancreatic ductal adenocarcinoma. PCN = pancreatic cystic neoplasms.

**Table 1 cancers-14-04228-t001:** Baseline and tumor characteristics.

	PCN-PDAC *n* = 233	PDAC *n* = 1761	*p*-Value
**Baseline characteristics**			
Male, *n*(%)	116 (50)	923 (52)	0.45
Age in years, mean (SD)	67 (9.5)	68 (8.8)	**0.024**
WHO performance status, *n* (%) ^a^			0.46
0–1	215 (92)	1599 (91)	
2–4	18 (8)	162 (9)	
(Neo)adjuvant therapy, *n* (%)			0.08 ^b^
Chemotherapy	128 (55)	1068 (61)	
(Chemo)radiotherapy	-	16 (0.9)	
**Tumor characteristics**			
pT-stage, *n* (%) ^c,d^			**<0.001**
T1	34 (15)	60 (3)	
T2	80 (34)	485 (27)	
T3	117 (50)	1151 (65)	
T4	2 (0.9)	65 (2)	
pN-stage, *n* (%) ^c,e^			**<0.001**
N0	100 (43)	475 (27)	
N1	111 (48)	1098 (62)	
N2	22 (9)	188 (11)	
R0 resection, *n* (%) ^f^	120 (52)	748 (42)	**0.009**
Vascular invasion, *n* (%) ^g^	102 (44)	1082 (61)	**<0.001**
Perineural invasion, *n* (%) ^h^	164 (73)	1523 (86)	**<0.001**

Percentages might not sum to 100% because of rounding. ^a^ WHO status was missing for 1156 (58%) of the patients in the original dataset. ^b^ Fisher’s exact test was used. ^c^ Depending on the time of registration, tumor and lymph node staging were based on either the seventh or eighth edition of the pathological tumor-node-metastasis (TNM) classification of malignant tumors. ^d^ T-stage was missing for 116 (5.8%) of the patients in the original dataset. ^e^ N-stage was missing for 12 (0.6%) of the patients in the original dataset. ^f^ Radical resection was missing in 5 (0.3%) of the patients in the original dataset. R0-2 was defined according to the International Collaboration on Cancer Reporting. ^g^ Vascular invasion was missing for 438 patients (22%) in the original dataset. ^h^ Perineural invasion was missing for 250 patients (13%) in the original dataset. Abbreviations: *n* = number. PCN = pancreatic cystic neoplasms. PDAC = pancreatic ductal adenocarcinoma. pT = tumor stage. pN = nodal stage. SD = standard deviation. WHO = World Health Organization. Bold means significant difference.

**Table 2 cancers-14-04228-t002:** Cox regression analyses evaluating factors associated with overall survival.

	Univariate Analysis			Multivariable Analysis		
	HR	95%CI	*p*-value	HR	95%CI	*p*-value
Female	0.96	0.87–1.06	0.38	-	-	-
Age (years)	1.01	1.01–1.02	**<0.001**	1.01	1.01–1.02	**<0.001**
WHO score	1.21	0.91–1.62	0.19	-	-	-
ASA score 3−4	1.20	1.05–1.37	**0.01**	1.16	1.03–1.32	**0.015**
PCN-PDAC	0.53	0.44–0.63	**<0.001**	0.65	0.55–0.78	**<0.001**
pT-stage ^a^						
T1	Ref			Ref		
T2	2.00	1.46–2.72	**<0.001**	1.11	0.81–1.53	0.52
T3	2.64	1.96–3.55	**<0.001**	1.32	0.98–1.78	0.07
T4	3.95	2.71–5.75	**<0.001**	1.71	1.14–2.56	**0.009**
pN-stage ^a^	2.07	1.84–2.32	**<0.001**	1.65	1.44–1.88	**<0.001**
Perineural invasion	1.98	1.66–2.35	**<0.001**	1.42	1.19–1.70	**<0.001**
Vascular invasion	1.67	1.49–1.88	**<0.001**	1.14	1.00–1.30	0.05
R0/1 resection ^b^	1.52	1.37–1.68	**<0.001**	1.25	1.13–1.39	**<0.001**
Differentiation grade						
Well differentiated	Ref			Ref		
Moderately differentiated	1.44	1.20–1.72	**<0.001**	1.28	1.07–1.52	**0.006**
Poorly differentiated	2.14	1.78–2.56	**<0.001**	1.83	1.51–2.21	**<0.001**

Abbreviations: ASA = American Society of Anesthesiologists. CI = confidence interval. HR = hazard ratio. PCN = pancreatic cystic neoplasms. PDAC = primary ductal adenocarcinoma. T = tumor. WHO = World Health Organization. ^a^ Depending on the time of registration, tumor and lymph node staging were based on either the seventh or eighth edition of the pathological tumor-node-metastasis (TNM) classification of malignant tumors. ^b^ R0-2 was defined according to the International Collaboration on Cancer Reporting. Bold means significant difference.

## Data Availability

The data that support the findings of this study are available from the corresponding author upon reasonable request. The data are not publicly available due to the use of individual patient data, which can only be shared after de-identification and approval by the study team. More importantly, a data transfer agreement has to be set up prior to data sharing.

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
