# Peer review of "Comparing Survival after Resection of Pancreatic Cancer with and without Pancreatic Cysts: Nationwide Registry-Based Study"

_cancers, 2022, doi:10.3390/cancers14174228_

Round 1

Reviewer 1 Report

Dr. Hooft and Gorris compared the overall survival of pancreatic cancers with or without pancreatic cysts. This study would be more helpful for readers, if the following points can be solved.

1. PCNs include 3 conditions in this manuscript. It would be better to also separately analyze each condition as well. 

2. Did the size of PCNs have any effects on the survival.

3. For cases with R0 resection, is there any difference in terms of disease-free interval?

4.  Are there any differences on the presenting symptom & signs? How about the cachexia incidence in cases with and without PCNs?

5. How about recurrent patterns or causes of death in these conditions?

Reviewer 2 Report

I salute the authors on a well written manuscript verifying that surgery for cyst-related malignancies are better off after treatment than pure PDACs.

I would like to see comments in the discussion how this can be. Do we think that adenocarcinoma connected to a cyst (or not at all connected to, but within the same organ) is less deadly than solid PDAC?

Often, cystic lesions are found by accident (patient asymptomatic). Cancer prophylactic resection are recommended (sometimes after years of surveillance). Adenocarcinoma in early stage is found by surprise. On the other hand, solid PDACs are found after investigation of jaundice, weight-loss, fatigue etc., factors that may affect survival?

Early stage PCL-PDACs are easier to resect than T3-4 PDACs. More Whipple-procedures in PDAC group. What about post-op complications? This must be mentioned when describing differences in OS.

The groups are defined solely by the pathology report. Did the annual number of reported PCL-PDACs change during the study-period, or did IPMN/MCN get more attention by the pathologists in the end of the period? Did the pathologists in all the hospitals follow the same standardized protocol for reporting of other findings than the PDAC?

As the authors mention, I think there are a not negligible number of cysts in the pancreatic remnant in the PDAC-group that would be put in the PCN-PDAC-group if they were known. Since no access to radiology, it might be reasonable to only chose adenocarcinoma with obvious connection to a cyst in the PCN-PDAC-group, and the rest in the PDAC-group (even PDACs concomitant with PCNs elsewhere)?

The authors could maybe expand the section of future perspective? What is the next step? Only prospective OS-trials, or biomarker/tumor biology studies?

Round 2

Reviewer 1 Report

No more questions